# Vascular Function in Continuous Flow LVADs: Implications for Clinical Practice

**DOI:** 10.3390/biomedicines11030757

**Published:** 2023-03-02

**Authors:** Fouad Khalil, Rabea Asleh, Radha Kanneganti Perue, Jean-Marc Weinstein, Adam Solomon, Batya Betesh-Abay, Alexandros Briasoulis, Hilmi Alnsasra

**Affiliations:** 1Department of Internal Medicine, University of South Dakota, Sioux Falls, SD 57105, USA; 2Heart Institute, Hadassah University Medical Center, Faculty of Medicine, Hebrew University of Jerusalem, Jerusalem P.O. Box 12000, Israel; 3Department of Cardiovascular Medicine, Mayo Clinic, Rochester, MN 55902, USA; 4Department of Cardiovascular Medicine, University of Nebraska Medical Center, Omaha, NE 68198, USA; 5Faculty of Health Sciences, Ben Gurion University of the Negev, Beersheva P.O. Box 653, Israel; 6Department of Cardiology, Soroka University Medical Center, Rager Av., Beersheva P.O. Box 84101, Israel; 7Department of Cardiovascular Medicine, University of Iowa Hospitals and Clinics, Iowa City, IA 52242, USA

**Keywords:** continuous flow left ventricular assist device, gastrointestinal bleeding, stroke, pulmonary hypertension, peripheral artery disease, coronary artery disease

## Abstract

Left ventricular assist devices (LVADs) have been increasingly used in patients with advanced heart failure, either as a destination therapy or as a bridge to heart transplant. Continuous flow (CF) LVADs have revolutionized advanced heart failure treatment. However, significant vascular pathology and complications have been linked to their use. While the newer CF-LVAD generations have led to a reduction in some vascular complications such as stroke, no major improvement was noticed in the rate of other vascular complications such as gastrointestinal bleeding. This review attempts to provide a comprehensive summary of the effects of CF-LVAD on vasculature, including pathophysiology, clinical implications, and future directions.

## 1. Introduction

Left ventricular assist device (LVAD) support has become a valuable therapeutic option to improve survival and quality of life in patients with advanced heart failure [1]. Despite the advancements in LVAD design, long-term exposure to continuous-flow (CF) LVADs has been linked to vascular dysfunction and major vascular sequelae, including bleeding and thrombosis. The endothelium plays a major role in vascular dysfunction exhibited by CF-LVAD recipients. The reduced pulsatility in CF-LVADs is thought to be a major factor for endothelial dysfunction. Moreover, supraphysiological shear stress in CF-LVADs results in hemolysis, von Willebrand factor (VWF) degradation and other changes that eventually contribute to the development of vascular pathology [2,3,4,5]. The non-physiological flow pattern in CF-LVADs worsens the endothelial dysfunction and results in elevated reactive oxygen species, generation of proinflammatory factors, platelet activation, vascular wall permeability, dysregulated vascular tone, and nitric oxide (NO) deficiency [6]. These changes in the vascular bed lead to several vascular complications that can impact the quality of life, heart transplant (HT) probability, and survival in CF-LVAD patients.

In this review, we summarize the effects of CF-LVADs on vascular function in the context of the clinical burden of vascular consequences, together with strategies to minimize the risk of these consequences. Moreover, we highlight important areas of future research that will advance our understanding of the physiology of this patient population and help to develop novel management strategies for vascular complications in CF-LVAD patients.

## 2. The Impact of CF-LVAD on Brain Vessels and Neurologic Events

Although there are no studies directly evaluating cerebral blood flow (CBF) before and after LVAD implantation, Cornwell et al. found that middle cerebral arterial velocity, among both CF-LVAD and pulsatile LVAD patients, was comparable to healthy controls [7]. The reduction of pulsatility in CF-LVAD-supported patients leads to unloading of the arterial baroreceptors with a subsequent increase in neurohumoral activation and muscle sympathetic nerve activity [8,9]. The net effect of the sympathetic overdrive is an elevation in mean arterial pressure and reduction in pulse pressure, mainly due to an increase in the diastolic pressure.

Because of the predisposition to uncontrolled blood pressure (BP) in CF-LVAD patients, chronic hypertension leads to a rightward shift of the autoregulatory curve and a reduction in maximal dilator capacity of the cerebral vasculature [10]. However, two previous studies have demonstrated that cerebral autoregulation is normal among CF-LVAD patients; these data reinforce the importance of BP control in this population to minimize the risk of adverse cerebrovascular events [7,11].

Among patients supported with earlier devices, including both pulsatile and CF-LVADs, a high prevalence of micro-embolic signals was found using transcranial Doppler [12,13]. It was suggested that micro-emboli from pulsatile pumps were solid, whereas those from CF-LVADs were predominantly gaseous [14]. Autopsy studies show an extremely high prevalence (up to 90%) of cerebrovascular pathology, including hemorrhage and infarcts in CF-LVAD patients [15,16].

With the steady increase in the utilization of LVADs and the increase in duration of LVAD support in many patients, neurological injury remains the leading cause of death in these patients, specifically ischemic and hemorrhagic stroke [17]. While ischemic strokes represent 85% of all strokes in the general population, the occurrence of hemorrhagic and ischemic strokes seems similar in LVAD patients [18]. The pathophysiology of stroke in these patients remains poorly understood. However, LVAD-related factors ascribed to non-pulsatile flow have been implicated, including endothelial dysregulation, reduced NO bioavailability, and vascular smooth muscle proliferation, which may impair cerebral autoregulation and predispose these patients to ischemic and hemorrhagic strokes (Figure 1). Moreover, the acquired VWF deficiency was previously found to be associated with increased risk of hemorrhagic stroke [19]. Additionally, LVAD-related infection has been associated with increased risk of hemorrhagic and ischemic strokes. Presumed mechanisms include increased inflammation and oxidative stress, septic emboli, high rates of hemorrhagic transformation of ischemic strokes, and mycotic aneurysm formation [20]. Hypertension is a major risk factor for stroke among LVAD patients. The ENDURANCE Supplemental trial, which utilized enhanced BP protocol (MAP ≤ 85 mmHg), showed significantly lower hemorrhagic stroke events among HeartWare ventricular assist device (HVAD) (Medtronic, Minneapolis, MN, USA) patients compared to the prior ENDURANCE trial, where BP was less strictly controlled [21,22].

The third-generation LVADs also appear superior to the second-generation LVADs in terms of stroke-free survival. A randomized control trial showed a significantly higher incidence of stroke in the HeartMate II (HM2, Abbott Labs; Lake Bluff, IL, USA) control group (29.7%) versus the HeartWare study group (12.1%) [22], and data from the MOMENTUM 3 trial showed a significantly improved stroke-free survival at 2 years with HeartMate III (HM3, Abbott Labs; Lake Bluff, IL, USA) (78%) versus HM II (56%) [23]. This is likely secondary to better pump design with less shear stress, as supported by a study that demonstrated greater preservation of VWF structure in HM3 compared to HM2 [24].

Challenges in the interpretation of standard stroke imaging modalities are considered a major impediment to our understanding of the mechanisms underlying LVAD-related stroke. Patients with CF-LVAD might appear to have slower cerebral blood flow on computed tomography angiography (CTA), which can be confused with poor collateral circulation [25]. Multiphasic CTA can be a helpful tool to overcome poor vessel opacification by providing better temporal resolution compared to single-phase CTA [26]. Furthermore, the presence of magnets and metal in the current CF-LVADs preclude the use of magnetic resonance imaging (MRI) for stroke evaluation. This adds another layer of complexity in evaluation since small strokes, hemorrhagic conversion, and microhemorrhages can be missed [27].

### Pharmacological Strategies to Minimize the Risk of Ischemic and Hemorrhagic Stroke

Antiplatelets and anticoagulation with vitamin K antagonists are currently standard for anti-thrombotic prophylaxis in CF-LVAD. Careful monitoring and control of anticoagulation to balance thrombosis and bleeding risks, as well as good BP control with a mean arterial pressure goal <90 mmHg, are mainstays in primary and secondary stroke prevention in CF-LVAD-implanted patients [28]. For hemorrhagic strokes, the decision to reverse anticoagulation should be weighed against the risk of pump thrombosis. This is more feasible and safer now with the use of HM3 pumps due to the extremely low risk of pump thrombosis. In cases of hemorrhagic transformation of ischemic strokes, anticoagulation reversal may increase the risk of additional thrombosis.

Phosphodiesterase 5 inhibitors (PDE-5is) are well known to have antiplatelet and antithrombotic properties, in addition to hemodynamic benefits for right ventricular (RV) unloading via NO-mediated vasodilation. A large retrospective analysis using INTERMACS registry data has found that PDE-5i use after CF-LVAD implantation was associated with reduced risk of LVAD thrombosis, ischemic stroke, and all-cause mortality in both centrifugal and axial flow devices over a 2-year study period [29]. There was no difference in hemorrhagic stroke risk. The tradeoff, however, was an increased risk of gastrointestinal bleeding (GIB).

## 3. Impact of CF-LVAD on Gastrointestinal Vasculature and GIB

VWF plays a vital role in pathophysiology of GIB (Figure 2). The attenuated pulsatility in CF-LVAD seems to result in direct inhibition of VWF secretion by endothelial cells and indirect inhibition by reducing endothelial NO secretion that leads to negative-feedback inhibition of VWF secretion [30].

VWF deficiency in the context of CF-LVAD occurs either through pump shear-induced VWF activation with subsequent exposure of the ADAMTS-13 cleavage site for enzymatic degradation or by shear stress-induced fragmentation of VWF into dysfunctional small fragments, independent of ADAMTS-13 [4,31,32,33,34]. The net effect of decreased production and increased degradation of VWF culminates in diminished VWF-dependent platelet aggregation.

The lack of physiological pulsatility in CF-LVAD-implanted patients results in GI mucosal hypoxia with subsequent sympathetic activation and release of angiogenesis factors such as VEGF and angiopoietin 2. These factors cause smooth muscle relaxation and dilation of the mucosal veins, which results in arteriovenous malformations (AVM) and angiodysplasias [35,36]. VWF is thought to be a negative regulator of angiogenesis by reducing VEGF-2-dependent proliferation of endothelial cells by extracellular binding to integrin αvβ3 [37]. The loss of VWF with subsequent defective Weibel Palade body formation promotes angiodysplasia due to ineffective intracellular storage and release of angiopoietin-2 [37,38].

Notably, a contemporary study has suggested that VWF becomes hyper-adhesive in CF-LVAD patients rather than being excessively cleaved [39]. The hyper-adhesive VWF was shown to activate platelets and produce platelet-derived extracellular vesicles with subsequent local concentration of VEGF and development of aberrant angiogenesis [39].

### 3.1. Clinical Perspective

Despite the beneficial role of LVADs in patients with advanced heart failure, GIB is one of the major complications in this patient population [40,41]. The reported incidence of GIB post-CF-LVAD implantation ranges from 21% to 31% [42,43]. Multiple studies of GIB in LVADs reported that the majority of cases had therapeutic or subtherapeutic INR levels at the time of bleeding [43]. Predictors of GIB include older age, redo sternotomy, preoperative inotrope use, elevated preoperative creatinine, RV failure, and concomitant antiplatelet and anticoagulant use [42].

Upper GIB seems to be the main location of bleeding in LVAD patients with AVM, with angiodysplasia being the most common culprit [43,44]. In a pooled analysis of 1087 patients, the mean duration from CF-LVAD implantation to first bleeding event was 54 days, and anemia was the most common presentation, followed by melena [45].

The occurrence of GIB is associated with increased morbidity and mortality. Moreover, the need for repeated blood transfusions increases alloimmunization risk, which may limit HT offers [46]. This prompts the need to develop treatment strategies to prevent GIB.

### 3.2. Interventions and Medications to Reduce GIB in CF-LVAD-Implanted Patients

#### 3.2.1. Pump Design

The HM 3 device is recognized as a fully magnetically levitated device that has the potential to reduce shear stress, and it provides artificial pulsatility. Notwithstanding, these design changes did not lead to lower GIB when compared to HM 2 in the momentum trial [47]. Netuka et al. documented an 18% decrease in VWF with the HM 3 device, compared with a 46% to 73% reduction with the HM II device, after 45 days of support, with no measurable differences in ADAMTS-13 activity levels [48]. This suggests that the HM 3 device may still induce mechanical shear stress adequate to disturb VWF homeostasis. Furthermore, it seems that the HM 3 device likely provides an arterial pulsatility below the physiologic minimum level needed to reduce bleeding events.

The markedly lower pump thrombosis rates with newer CF-LVADs, particularly the HM 3 LVAD, has prompted discussion regarding the potential for avoidance of antiplatelet therapy to reduce bleeding risk based on observation data showing lower risk of re-bleeding without increase in thrombotic complications after discontinuation of aspirin in HM2 and HM3 LVAD patients [49,50,51,52]. However, the feasibility of such a strategy may be device-specific, as an Aspirin dose of 81 mg daily instead of 325 mg daily was associated with increased risk of thrombosis for the HeartWare HVAD pump, which was not the case for the HM3 and HM 2 devices [53,54,55,56].

#### 3.2.2. Medical Management

There is limited data regarding the potential benefit of pharmacological interventions, most of which comes from observational studies in patients with recurrent or refractory LVAD-related GIB.

It has been hypothesized that angiotensin-converting enzyme inhibitors (ACEi)/angiotensin receptor blockers (ARBs) may reduce angiogenesis by inhibiting angiotensin II-related activation of the transforming growth factor beta (TGF-β) and VEGF pathways. A recent systematic review and meta-analysis by Kittipibul et al. found that retrospective data from 3 studies [57,58,59] with a total of 619 CF-LVAD patients showed ACEi/ARB use was associated with a decreased incidence of overall GIB [60]. Interestingly, while there was a trend towards reduced odds of AVM-related GIB with ACEI/ARB use, it was not statistically significant as the largest study by Shultz et al. with 377 patients found no significant difference in AVM-related GIB rates by ACEi/ARB usage [58]. The protective effect seems to be seen with a dose threshold of >5 mg daily lisinopril equivalence rather than being dose-dependent [59] and seems independent of BP effect [57]. Unfortunately, the definitions of ACEI/ARB usage and GIB events vary amongst these studies. Furthermore, there is conflicting data showing no significant association between GIB risk and ACEi/ARB use in a retrospective analysis using data from 13,732 patients in the INTERMACS registry, about 52% of whom were on ACEi/ARB [61]. Interestingly, this analysis showed a lower risk of GIB in patients on beta blockers [61], which was not the case in the smaller study by Houston et al. [57].

The anti-VEGF monoclonal antibody, bevacizumab, was well tolerated and markedly reduced the need for transfusions, endoscopies, and GIB-related hospitalizations in a small pilot study involving five HM II LVAD patients with refractory angiodysplasia-related GIB over a median follow-up period of 22 months [62].

The somatostatin analog, octreotide, lowers portal pressures by splanchnic vasoconstriction and downregulates VEGF and basic fibroblast growth factor to inhibit angiogenesis, and it has been used in variceal bleeding as well as non-variceal angiodysplasia-related GIB [63]. It is well tolerated, with most of the available data from small observational studies showing some potential benefit in reducing GIB recurrence in CF-LVAD patients [63,64,65,66,67,68,69,70].

The data regarding digoxin’s potential role in LVAD-related GIB management is inconclusive. There are a few retrospective studies linking digoxin use to significant reduction in all-cause GIB, particularly in angiodysplasia-related GIB in CF-LVAD patients [71,72,73]. The proposed mechanism is suppression of hypoxia-inducible factor-1 α (HIF-1α), a mediator of angiopoietin-2-induced angiodysplasia [71]. However, a large retrospective analysis using the INTERMACS database by Jennings et al. in 2020, with over 2000 CF-LVAD patients on digoxin, found no association between LVAD-related GIB rates and digoxin use [61].

There is very limited data regarding the use of hormone therapy in CF-LVAD-related GIB. A small single-center proof-of-concept retrospective observational study has suggested significant reduction in GIB-related transfusions and hospitalizations with danazol use [74]. There is conflicting data regarding estrogen-based hormone therapy for AVM-related GIB prevention, and the potential for increased thromboembolic risk poses reservations.

Thalidomide is thought to downregulate HIF-1α expression and inhibit VEGF and basic fibroblast factor [75,76,77,78]. Its antiangiogenic properties have shown some promise in refractory angiodysplasia-related GIB, including in LVAD patients [78,79,80,81,82]. The largest retrospective study to date showed that thalidomide use in 17 CF-LVAD patients with angiodysplasia-related GIB was found to significantly reduce the risk of rebleed, median number of GIBs per year, and transfusion requirements per year while on thalidomide versus while off thalidomide (before initiation) [78]. Adverse event rate was 59%, albeit with dose reduction resolving symptoms in most patients without increased GIB [78]. Barriers to its use include high incidence of adverse effects which seem to be dose related, with unclear minimal effective dose, and the need for provider and pharmacist enrollment in the THALOMID Risk Evaluation and Mitigation Strategy (REMS) program to prescribe thalidomide due to teratogenicity.

Desmopressin is a vasopressin analog currently used to treat Hemophilia A and von Willibrand disease. It shortens bleeding time and improves hemostasis by increasing VWF and factor VIII levels, making it an attractive potential therapy for the acquired VWF deficiency implicated in LVAD-related GIB. In one case report, desmopressin prevented rebleeding for 6 months in one HM II LVAD patient with refractory GIB, despite holding antithrombotic therapies and starting octreotide [83]. The data is inadequate to provide a recommendation, as further studies are needed to determine efficacy and safety given potential for hyponatremia, fluid retention, and thrombosis.

## 4. The Effects of CF-LVAD on Pulmonary Vasculature and Pulmonary Hypertension

Chronic elevation of left-sided filling pressures is thought to trigger pulmonary vascular pathology through mechanical stress in the pulmonary venous system, leading to enhanced endothelin-1 expression, decreased NO availability, and subsequent arterial remodeling that leads to pulmonary hypertension (PH) [84]. If untreated, PH can cause pulmonary vasoconstriction and further arterial wall remodeling, which entails medial hypertrophy and intimal fibrosis. These changes alter the hemodynamics of pulmonary vasculature and increase pulmonary vascular resistance (PVR) [85,86].

LVAD implantation can reduce PVR by unloading the left ventricle (LV), reducing filling pressures, and augmenting cardiac output (Figure 3). However, the augmentation of systemic venous return can increase RV preload. As the precapillary component of PH can persist after normalization of the LV filling pressures in those with combined post-capillary and pre-capillary pulmonary hypertension, the RV also faces a pulmonary vasculature that is less compliant and has greater resistance. The combination of preload stress and increased afterload increases the risk of clinical RV failure and the attendant consequences, including persistent heart failure, hepatic congestion, GIB, renal dysfunction, and increased mortality [84].

### 4.1. Clinical Perspective

RV failure occurs in 10 to 40% of LVAD patients and represents an important cause of morbidity and mortality [87,88,89,90,91]. It is well known that some patients continue to have elevated PVR after LVAD implantation [92,93,94]. The reduced pulmonary arterial compliance associated with high PVR impairs RV output and subsequently precipitates systemic congestion in the settings of the high RV preload following LVAD implantation. Therefore, PVR has been suggested as a predictor of LVAD-associated RV failure in several studies [95,96,97,98].

The diastolic pulmonary gradient (DPG) between diastolic pulmonary artery pressure (dPAP) and mean pulmonary capillary wedge pressure (PCWP) was previously suggested as an index of pulmonary vascular remodeling, and a cutoff of 7 mm Hg has been previously proposed as a strong predictor of RV failure [99]. A previous study that investigated the effect of LVAD on DPG in 116 end-stage HF patients with PH-left heart disease (LHD) highlighted that despite the DPG decline after LVAD therapy, it remained significantly elevated (>7 mm Hg) in 42% of these patients [100]. DPG > 8 mm Hg was found to be significantly associated with nonresponse to LVAD therapy [100]. Moreover, two studies by Imamura et al. demonstrated that the DPG > 5 mm Hg at incremental LVAD speeds is associated with worse prognosis following LVAD implantation [101,102].

### 4.2. Transplant Considerations

High PVR is a well-established risk factor for negative outcomes post HT [103,104,105]. Therefore, PVR elevation of >5 Wood units (WU) or inability to reduce PVR < 2.5 WU with vasodilators are considered relative contraindications for HT as per the International Society for Heart and Lung Transplantation guidelines [103,106]. Implantation of LVADs in these patients allows reconsideration of HT in patients who were initially deemed ineligible as many studies have documented a reduction in the fixed PVR elevation post LVAD [107,108,109]. The mechanism by which LVAD reduces the fixed PVR elevation is not completely understood. Reduction of pulmonary pressures along with significant cardiac output improvement following LVAD implantation are thought to be major contributors to PVR reduction [107].

It might be speculated that continuous unloading of the LV reverses PH-induced pulmonary vasculature remodeling [109]. Although animal studies supported this theory by demonstrating a regression of PH-induced remodeling via hemodynamic unloading, human data are lacking [110].

An analysis of the INTERMACS registry found that PVR decreases mostly in the first 3 months following LVAD implantation and continues to decrease gradually but at a lower rate thereafter [111].

This initial reduction was attributed to the acute reduction in PCWP after implantation. The authors suggested that the gradual reduction after the 3 months is probably related to favorable structural remodeling of pulmonary vasculature due to lower pulmonary pressures [111].

Regardless of the mechanism, studies suggested that patients who had their elevated PVR normalized by LVAD had comparable post-HT outcomes to those without PH [112,113]. Conversely, Tsukashita et al. suggested higher mortality post HT in patients with pre-LVAD PVR of >5 WU, despite normalization of elevated PVR after LVAD implantation. The authors speculated the presence of other unknown PH indices that persist following LVAD implantation and affect HT outcomes [114].

### 4.3. Medical Treatment

RV-pulmonary arterial (PA) coupling refers to the interdependent relationship between RV contractility and afterload. RV-PA is considered to be “coupled” when contractility matches the afterload to maintain RV output. RV-PA uncoupling leads to decreased LV filling with subsequent suck-down events and ventricular arrhythmias in addition to systemic congestion and persistent HF symptoms [84].

LVAD patients frequently require inotropic and RV assist device (RVAD) support post LVAD implantation to maintain RV-PA coupling. The purpose of using pulmonary vasodilators in the early postoperative phase is to wean inotropic and RV device support. Multiple studies reported the safety and efficacy of pulmonary vasodilators such as milrinone, sildenafil and inhaled NO to reduce mPAP in the postoperative phase after LVAD implantation [115,116,117,118]. However, whether this can be extrapolated to long-term management is still unknown.

Although mounting evidence supports PVR reduction after LVAD implantation, there is a significant subset of patients who continue to have elevated PVR after LVAD [92,93,94]. Management of persistent PH in these patients is important to reduce RV dysfunction and enhance PVR reduction in preparation for HT. Although guidelines recommend PDE-5 inhibitor use in LVAD patients with persistent PH and RV dysfunction, there is no strong evidence to support the benefits of this approach [119]. Nevertheless, there is a growing body of evidence from observational and non-randomized studies suggesting good tolerability and possible beneficial effects of long-term use of pulmonary vasodilators to reduce PVR and RV failure [92,93,120]. Tedford et al. illustrated that the use of sildenafil reduces PVR in LVAD patients and allows bridging to transplant [93]. Another study showed good tolerability and PVR reduction with bosentan use in patients with persistent PH after LVAD implantation [92]. However, randomized trials are needed to prove the safety and efficacy of the long-term use of these medications in LVAD patients.

## 5. Peripheral Arterial Disease

CF-LVAD patients frequently have peripheral arterial disease (PAD) even prior to LVAD implantation due to similar risk factors between HF and PAD such as hypertension, diabetes, dyslipidemia, and smoking [121,122]. PAD might worsen after CF-LVAD implantation due to a further decline in peripheral vascular function secondary to endothelial dysfunction. The lack of pulsatility leads to decreased production of endothelial-derived vasodilatory substances such as NO. This results in worsening endothelial dysfunction and decreased peripheral perfusion [123]. Additionally, the persistent elevation of inflammatory mediators in HF patients after CF-LVAD implantation might contribute to a further decline in endothelial function [124,125]. Contact between blood and the artificial surface might further augment the pre-existing HF-related inflammatory process [124]. A study has shown higher levels of inflammatory mediators, such as granulocyte-macrophage colony-stimulating factor (GM-CSF), macrophage-derived chemokine (MDC), and macrophage inflammatory protein1-β (MIP-1β) after CF-LVAD implantation [124]. Furthermore, the non-pulsatile flow can upregulate the renin-aldosterone angiotensin system with a subsequent increase in inflammatory markers, such as interleukin-6 (IL-6), IL-8, tumor necrosis factor-α (TNF-α), monocyte chemoattractant protein-1 (MCP-1), and C-reactive protein (CRP) [126,127,128,129]. These inflammatory mediators were found to be implicated in vascular dysfunction, atherogenesis, increased macrophage accumulation, procoagulant state, and inhibition of antithrombotic proteins such as antithrombin and protein C [130,131,132,133,134].

### Clinical Perspective

While CF-LVAD can worsen PAD, preexisting PAD is associated with many negative outcomes following CF-LVAD implantation. A study of 20,817 patients with LVADs showed that patients with pre-existing PAD had higher risk of in-hospital mortality and complications compared to those without PAD [135]. PAD was found to increase late mortality in LVAD patients as well [136]. Therefore, PAD is considered a relative contraindication for LVAD implantation [103].

PAD can impact the risk of thrombosis and bleeding in LVAD patients. Studies have illustrated higher HAS-BLED scores in PAD patients [137]. Hence, LVAD patients with PAD may be at higher risk of major bleeding events [135].

In addition to increasing the risk of bleeding, PAD was found to increase the risk of thrombosis in CF-LVAD patients [135]. Indeed, bleeding in PAD patients was found to be an independent predictor of subsequent major ischemic events [138].

Evaluation of PAD in the context of CF-LVAD might be challenging. The reliability of ankle-brachial index is unproven in these patients [139]. The inadequate visualization of peripheral vasculature, along with the continuous blood flow in the context of CF-LVAD, limit the role of standard imaging modalities such as Doppler ultrasound. Notably, Falletta et al. have suggested that flow simulation with computational flow analysis might add to the yield of the current imaging tools in CF-LVAD patients [139].

Given the impact of CF-LVAD on peripheral vasculature, screening for PAD should be implemented in all patients undergoing evaluation for LVAD implantation, especially those with risk factors for developing PAD.

## 6. Coronary Arteries

The impact of CF-LVADs on the coronary arteries is poorly understood. It is thought that continuous flow in CF-LVAD creates abnormal coronary hemodynamics and impacts the arterial structure. Endothelial dysfunction due to loss of pulsatility is likely a major contributor to the pathophysiology. Patients with LVAD were found to have about a 33% increase in myocardial microvascular density measured by CD-34 staining [140]. Signs of endothelial activation/dysfunction in these patients include reduplication of basal lamina, increased cellular projections, and organelles in the laminal area [140]. The increase in microvascular density is likely secondary to activation of angiogenic pathways, including angiopoietin-2 signaling. Angiopoietin-2 pathway is frequently implicated in abnormal vascular growth such as AVM and mucosal bleeding [141]. Additional study has shown a significant breakdown of the coronary internal elastic lamina with thickening of the external elastic laminal in patients with CF-LVADs [142]. The breakdown of the elastin fibers leads to release of proteolytic products with subsequent chemotaxis of inflammatory cells and angiogenesis [143,144]. CF-LVAD patients were also found to have expansion of coronary adventitia with an increase in collagen deposition and vasa vasorum proliferation with subsequent fibrosis of coronary arteries [142].

### Clinical Implications

Although coronary arteries remodeling and fibrotic changes after CF-LVAD implantation could theoretically induce myocardial ischemia, the exact clinical impact of these changes remains unknown. In contrast, another study has shown that CF-LVAD support improves myocardial blood flow in a bovine model of chronic ischemic heart failure, most likely by increasing diastolic pressure [145]. Similarly, Symons et al. reported no change in coronary endothelial or vascular smooth muscle vasorelaxation (measured by response to bradykinin and nitroprusside, respectively) in 11 patients after 200 days following CF-LVAD implantation [146]. Furthermore, six patients with ischemic cardiomyopathy had improvement in coronary endothelial-dependent vasorelaxation following CF-LVAD support [146]. Therefore, further research is needed to identify the clinical impact of CF-LVAD support on the coronary arteries.

## 7. Future Directions

New pump designs are needed to further reduce bleeding and thrombosis with a view to increasing the number of LVAD recipients, and these pump designs should be aimed at mitigating bleeding and thrombosis. This could include pulsatility algorithms to mimic the physiologic pulsatile shear stress and to reduce mechanical shear.

Flow modulation strategies are currently being examined to generate pulsatility in centrifugal CF-LAVDs [147]. The HVAD speed-modulating function is being further developed to induce greater pulsatility [148]. The HM 3 produces near-physiologic pulse pressure; however, it can only generate pulse pressure of about 25 mm Hg [149]. Bozkurt et al. showed that it is possible to improve the pulsatility in CF-LVAD support by regulating pump speed over a cardiac cycle without compromising the overall level of support. To fulfill the perfusion requirements for different physiological conditions and enhance pulsatility, variable speed control systems have been developed. Several studies proposed the Frank-Starling mechanism to mimic the heart by using physiological feedback control systems that measure pressures and flow directly; or more recently, by using non-invasive estimation algorithms [150,151,152,153,154,155].

The HM3 device integrates a Full MagLev Flow Technology that maintains an improved hemocompatibility profile and provides artificial pulsatility. Despite the decreased incidence of pump thrombosis with HM 3 in the Momentum 3 clinical trial, there was no change in GIB (HM2: 27.3% vs. HM3 27.0%) [156]. The HM 3 device may still induce mechanical shear stress adequate to disturb VWF homeostasis, as suggested by Netuka et al. [48]. Following the abrupt withdrawal of the HeartWare device in June 2021 due to higher incidences of neurological events reported in several observational studies, HM3 became the only FDA-approved LVAD [157]. This has necessitated the need for innovations to create more options for advanced heart failure in the era of a single device.

The EVAHEART (EVAHEART Inc, Houston, TX, USA) is a pump designed to reduce VWF degradation through a series of features to increase pulsatility and reduce pump shear. Compared with the HM2, the EVAHEART has been reported to induce a smaller increase in the smallest VWF fragments in a mock circulatory loop with whole human blood [158]. In a post-market approval study in Japan, none of the 93 patients receiving the EVAHEART had GIB on follow-up, and no pumps were exchanged for pump thrombosis [159]. These data are encouraging and provide evidence that a multifactorial effort to increase arterial pulsatility and reduce pump shear have promise to diminish the clinically observed incidence of bleeding in LVAD recipients. The newer EVAHEART^®^2 (EVA2) is a smaller device with a new tipless inflow cannula design aimed at reducing stroke and other inflow cannula-related complications [160]. The ongoing randomized controlled COMPETENCE Trial will examine non-inferiority in safety and efficacy of the EVA2 versus the HM3 device [161].

The toroidal-flow TORVAD (Windmill Cardiovascular Systems, Inc., Austin, Texas) is a unique new device design proposition unlike any of the previous generations of membrane-type pulsatile or CF-LVADs. It sequentially spins two magnetic pistons within a donut-shaped “torus” chamber to simultaneously fill and eject blood in a unidirectional and pulsatile manner. TORVAD produces significantly lower shear stress due to operating at a relatively low speed (60–150 rpm). Furthermore, it provides physiological synchronized pumping through an integrated epicardial sensing lead to analyze heart rate and rhythm, and it senses and adjusts to changes in preload and afterload [162]. Unlike the HM2, TORVAD caused minimal VWF degradation and hemolysis and did not activate platelets in an ex vivo study, and sheep implanted with TORVAD showed similar findings with no thromboembolism despite lack of anticoagulation [162,163]. While it has not yet been tested in humans, it promises the potential for significant improvements in hemocompatibility.

In the era of the new-generation devices with improved hemocompatibility, reducing the antithrombotic regimen intensity is another area that should be targeted in order to reduce bleeding events. The ongoing international non-inferiority randomized control trial, Antiplatelet Removal and Hemocompatibility Events with the HM3 Pump (ARIES HM3), will test the hypothesis that foregoing aspirin as part of the antithrombotic regimen of HM3 LVAD patients would reduce risk of non-surgical bleeding without compromising safety and efficacy [53].

There has been interest in novel oral anticoagulants (NOACs) in LVAD patients. A randomized controlled trial of 30 HeartWare patients randomized to either dabigatran or vitamin K agonists was terminated early because of excess thromboembolic events in the dabigatran group [164]. A small retrospective study of 35 patients receiving the HM3 reviewed the use of warfarin versus apixaban (20 in the warfarin group and 15 in the apixaban group). At 6 months after implant, there was no statistically significant difference in death, stroke, bleeding, or other thrombotic complications between the warfarin and apixaban groups [165]. With caution and some optimism, the potential of apixaban in HM3 patients may need to be explored in a randomized trial.

With better understanding of the interplay between pump design innovation and improvements in clinical management, LVAD-related adverse events might be minimized. As novel engineering features in the HM3 have led to significant reductions in the incidence of pump thrombosis, it opens up an array of potential ways to fine-tune and study innovative anticoagulation or device-management strategies [166].

## 8. Conclusions

Despite the improvement in CF-LVAD devices, vascular complications remain a major concern that can impact quality of life and survival. Reduced pulsatility and increased mechanical shear stress appear to be the main factors implicated in the development of vascular pathology in CF-LVAD patients. Acquired VWF seems to be a common pathway for the development of vascular complications in CF-LVAD-supported patients. New device designs with better hemocompatibility, improved pulsatility, and less mechanical shear stress are needed to mitigate the CF-LVAD-associated vascular complications.

## Figures and Tables

**Figure 1 biomedicines-11-00757-f001:**
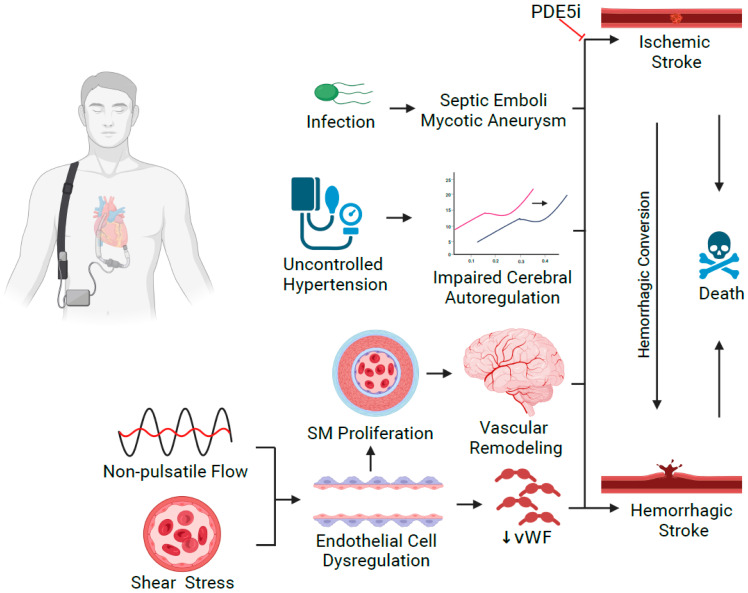
Pathophysiology of stroke in patients supported with CF-LVAD. Continuous flow and shear stress are the main contributors to the development of cerebrovascular pathology, while infection and hypertension are important risk factors. PDE5i: phosphodiesterase 5 inhibitors; SM: smooth muscle; VWF: von Willebrand Factor.

**Figure 2 biomedicines-11-00757-f002:**
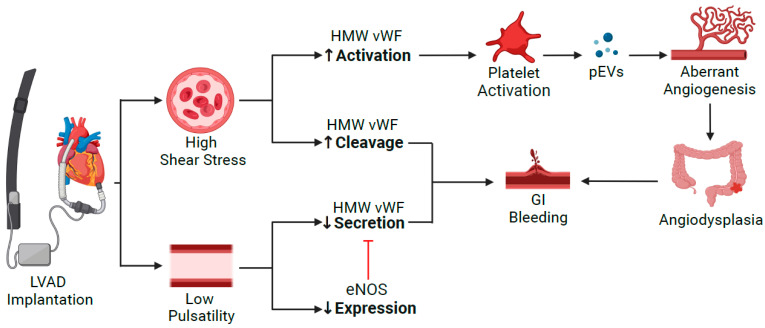
Pathophysiology of GIB in patients supported with CF-LVAD. VWF deficiency is the common pathway for development of angiodysplasia due to the high shear stress and continuous flow. eNOS: endothelial nitric oxide synthase; GI: gastrointestinal; HMW VWF: high molecular weight von Willebrand Factor; pEVs: platelet-derived extracellular vesicles.

**Figure 3 biomedicines-11-00757-f003:**
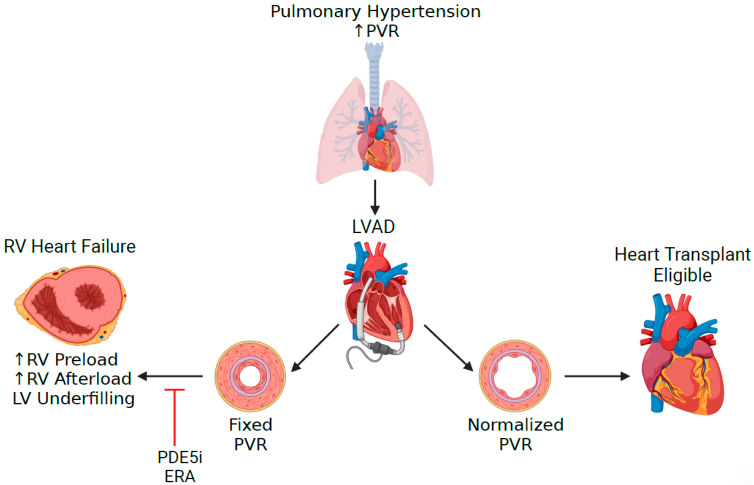
LVAD effect on pulmonary vasculature. Reduction in PVR increases heart transplant probability while persistent PVR elevation is associated with RV failure. ERA: endothelin receptor antagonists; LV: left ventricle; LVAD: left ventricular assist device; PDE5i: phosphodiesterase 5 inhibitors; PVR: pulmonary vascular resistance; RV: right ventricle.

## Data Availability

No new data were created or analyzed in this study. Data sharing is not applicable to this article.

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
