# Peer review of "Vascular Function in Continuous Flow LVADs: Implications for Clinical Practice"

_biomedicines, 2023, doi:10.3390/biomedicines11030757_

Round 1

Reviewer 1 Report

It is certainly a comprehensive and very well written review on the subject.

It would be worth mentioning the disappointing withdrawal of the HeartWare from the market in June 2021 due to unexpectedly high rate of neurological events and malfunction in terms of failure to restart. I still consider it a very good device. The HeartMate II remains a good device too despite criticism.

I would also suggest expanding a bit more on speed modulation (speed control and Frank-Starling controller) in the “Future Direction” section. Speed modulation is likely to play a significant role in mimicking pulsatility.

A list of abbreviations used throughout the manuscript would be helpful.

Here are some other comments and suggestions.

Line 29: it would be more appropriate to say “Left ventricular assist device (LVAD) support has become a valuable therapeutic option to improve survival and quality of life in patients with advanced heart failure.”

Line 44: it would be more appropriate to say “...in the context of the clinical burden of vascular complications...”

Line 67-68: it would be more appropriate to say “...whereas those from CF-LVADs were predominantly gaseous.”

Line 343: it would be more appropriate to say “...and enhance PVR reduction...”

Line 348: replace with “The use of sildenafil reduces PVR in LVAD patients...”

Line 416: it would be more appropriate to say “In contrast, another study has shown that CF-LVAD support improves myocardial blood flow in a bovine model of chronic ischaemic heart failure most likely by increasing the diastolic pressure.”

Line 424: replace with “...on the coronary arteries.”

Line 426: it would be more appropriate to rearrange as “New pump design are needed to further reduce bleeding and thrombosis with a view to increase the number of LVAD recipients.”

Line 480: it would be more appropriate to say “A small retrospective study of 35 patients receiving the HM3 reviewed the use of Warfarin versus Apixaban (...).”

Line 493: it is better to say “quality of life”

Author Response

Reviewer #1 Comments  

Reviewer 1 

It is certainly a comprehensive and very well written review on the subject. 

Comment 1: It would be worth mentioning the disappointing withdrawal of the HeartWare from the market in June 2021 due to unexpectedly high rate of neurological events and malfunction in terms of failure to restart. I still consider it a very good device. The HeartMate II remains a good device too despite criticism. 

Response:

We thank the reviewer for the constructive comments. We agree with the reviewer that the unfortunate withdrawal of HeartWare device has led to a practice dominated by a single device (HM3). We have updated our manuscript to mentions this point as can be seen on page 11, line 462 which states the following:

“Following the abrupt withdrawal of HeartWare device in June 2021 due to higher incidences of neurological events reported on several observational studies, HM3 became the only FDA-approved LVAD. [161] This has necessitated the need for innovations to create more options for advanced heart failure in the era of single device.”

Comment 2: I would also suggest expanding a bit more on speed modulation (speed control and Frank-Starling controller) in the “Future Direction” section. Speed modulation is likely to play a significant role in mimicking pulsatility. 

Response:

We thank the reviewer for raising this important point. We have expanded on this point in our discussion of future directions. However, we tried to avoid detailed engineering/mechanical aspects to target broader section of clinicians and to keep this comprehensive manuscript within acceptable size. We have updated our manuscript to include this as can be seen on page 10, line 444 which states:

“To fulfill the perfusion requirements for different physiological conditions and enhance pulsatility, variable speed control systems have been developed. Several studies proposed the Frank-Starling mechanism to mimic natural heart by using physiological feedback control systems that measure pressures and flow directly or more recently, by non-invasive estimation algorithms. [154–159]”

Comment 3: A list of abbreviations used throughout the manuscript would be helpful. 

Response:

We thank the reviewer for the opportunity to provide a complete list of abbreviations. We have updated the manuscript and included a comprehensive list of abbreviations at the end of manuscript.

Comment 4:

Here are some other comments and suggestions. 

 -Line 29: it would be more appropriate to say “Left ventricular assist device (LVAD) support has become a valuable therapeutic option to improve survival and quality of life in patients with advanced heart failure.” 

 -Line 44: it would be more appropriate to say “...in the context of the clinical burden of vascular complications...” 

 -Line 67-68: it would be more appropriate to say “...whereas those from CF-LVADs were predominantly gaseous.” 

 -Line 343: it would be more appropriate to say “...and enhance PVR reduction...” 

 -Line 348: replace with “The use of sildenafil reduces PVR in LVAD patients...” 

 -Line 416: it would be more appropriate to say “In contrast, another study has shown that CF-LVAD support improves myocardial blood flow in a bovine model of chronic ischaemic heart failure most likely by increasing the diastolic pressure.” 

 -Line 424: replace with “...on the coronary arteries.” 

 -Line 426: it would be more appropriate to rearrange as “New pump design are needed to further reduce bleeding and thrombosis with a view to increase the number of LVAD recipients.” 

-Line 480: it would be more appropriate to say “A small retrospective study of 35 patients receiving the HM3 reviewed the use of Warfarin versus Apixaban (...).” 

-Line 493: it is better to say “quality of life” 

 Response:

We thank the reviewer for these suggestions. We have updated the manuscript and applied all the corrections/suggestions.

 We thank the reviewer for their valuable comments.  We hope that we have addressed them satisfactorily and believe our manuscript has been strengthened as a result. 

Reviewer 2 Report

The authors described a comprehensive summary of the effects of CF-LVAD on vasculature. I think the authors made thorough review on this topic. I have several suggestions.

- In Introduction section, the term of "Vascular complication" is used multiple times. This term makes readers think about vascular injury. I think it does not reflect the theme of this review. It should be changed to another word. 

- P8, L307; "CO-improvement," what does CO mean?

- P10, L445; "HMV VWF degradation," what does HMV mean?

- There is an incomplete list of abbreviations at the end of the manuscript. A complete list of abbreviations should be added.

Author Response

Reviewer # 2 

The authors described a comprehensive summary of the effects of CF-LVAD on vasculature. I think the authors made thorough review on this topic. I have several suggestions. 

Comment 1: In Introduction section, the term of "Vascular complication" is used multiple times. This term makes readers think about vascular injury. I think it does not reflect the theme of this review. It should be changed to another word.  

Response:

We thank the reviewer for the constructive comments. We have updated the introduction section and replaced the term “complications” with “sequelae” and “consequences” as can be seen on page 1, lines 32, 45 and 46 which states:

“LVADs has been linked to vascular dysfunction and major vascular sequelae including bleeding and thrombosis”

In this review, we summarize the effects of CF-LVADs on vascular function in the context of the clinical burden of vascular consequences, placed together with strategies to minimize the risk of these consequences.

Comment 2: P8, L307; "CO-improvement," what does CO mean? 

Response:

We have replaced “CO” with “cardiac output” as can be seen on page 8 line 314:

“Reduction of pulmonary pressures along with significant cardiac output improvement following LVAD implantation are thought to be major contributors to PVR reduction”

Comment 3: P10, L445; "HMV VWF degradation," what does HMV mean? 

Response:

HMW stands for high molecular weight. We have deleted “HMW” as we felt it was redundant and we used “VWF degradation” to be consistent throughout the manuscript.

- Comment 4: There is an incomplete list of abbreviations at the end of the manuscript. A complete list of abbreviations should be added. 

Response:

We thank the reviewer for the opportunity to provide a complete list of abbreviations. We have updated the manuscript and included a comprehensive list of abbreviations at the end of manuscript

We thank the reviewer for their valuable comments.  We hope that we have addressed them satisfactorily and believe our manuscript has been strengthened as a result.